# Screening of Exosome-Derived Proteins and Their Potential as Biomarkers in Diagnostic and Prognostic for Pancreatic Cancer

**DOI:** 10.3390/ijms241612604

**Published:** 2023-08-09

**Authors:** Anelis Maria Marin, Michel Batista, Alexandre Luiz Korte de Azevedo, Talita Helen Bombardelli Gomig, Rodrigo Soares Caldeira Brant, Roger Chammas, Miyuki Uno, Diogo Dias Araújo, Dalila Luciola Zanette, Mateus Nóbrega Aoki

**Affiliations:** 1Laboratory for Applied Science and Technology in Health, Carlos Chagas Institute, Oswaldo Cruz Foundation (Fiocruz), Curitiba 81350-010, Brazil; anelis.marin@fiocruz.br (A.M.M.); michel.batista@fiocruz.br (M.B.); dalila.zanette@fiocruz.br (D.L.Z.); 2Mass Spectrometry Facility RPT02H, Carlos Chagas Institute, Oswaldo Cruz Foundation (Fiocruz), Curitiba 81350-010, Brazil; 3Laboratory of Human Cytogenetics and Oncogenetics, Genetic Department, University of Parana State (UFPR), Curitiba 80060-000, Brazil; alx.bio10@gmail.com (A.L.K.d.A.); talitahbg@ufpr.br (T.H.B.G.); 4Center for Translational Research in Oncology (LIM24), Instituto do Cancer do Estado de Sao Paulo (ICESP), Hospital das Clinicas da Faculdade de Medicina da Universidade de Sao Paulo (HCFMUSP), Comprehensive Center for Precision Oncology (C2PO), Universidade de São Paulo, São Paulo 05508-220, Brazil; rchammas@usp.br (R.C.); miyuki.uno@hc.fm.usp.br (M.U.); d.araujo@hc.fm.usp.br (D.D.A.)

**Keywords:** pancreatic cancer, exosome, plasma, proteomics

## Abstract

In the oncological area, pancreatic cancer is one of the most lethal diseases, with 5-year survival rising just 10% in high-development countries. This disease is genetically characterized by *KRAS* as a driven mutation followed by *SMAD4*, *CDKN2*, and *TP53*-associated mutations. In clinical aspects, pancreatic cancer presents unspecific clinical symptoms with the absence of screening and early plasmatic biomarker, being that CA19-9 is the unique plasmatic biomarker having specificity and sensitivity limitations. We analyzed the plasmatic exosome proteomic profile of 23 patients with pancreatic cancer and 10 healthy controls by using Nanoscale liquid chromatography coupled to tandem mass spectrometry (NanoLC-MS/MS). The pancreatic cancer patients were subdivided into IPMN and PDAC. Our findings show 33, 34, and 7 differentially expressed proteins when comparing the IPMN vs. control, PDAC-No treatment vs. control, and PDAC-No treatment vs. IPMN groups, highlighting proteins of the complement system and coagulation, such as C3, APOB, and SERPINA. Additionally, PDAC with no treatment showed 11 differentially expressed proteins when compared to Folfirinox neoadjuvant therapy or Gemcitabine adjuvant therapy. So here, we found plasmatic exosome-derived differentially expressed proteins among cancer patients (IPMN, PDAC) when comparing with healthy controls, which could represent alternative biomarkers for diagnostic and prognostic evaluation, supporting further scientific and clinical studies on pancreatic cancer.

## 1. Introduction

Pancreatic cancer is the third leading cause of cancer deaths in the United States [1] and is projected to become the second leading cause by 2030 [2]. With 495,773 new cases in 2020, pancreatic cancer represents the 12th disease in incidence worldwide; however, with 466,003 cases it is the 7th in the United States and represents 4.7% in cancer global death, being the third leading contributor to cancer mortality in the United States [1,3]. Worldwide, the incidence (5.7 and 4.1 per 100,000, respectively) and mortality (4.9 and 4.5 per 100,000, respectively) are higher in males than in females and correlate with increasing age [3]. Pancreatic adenocarcinoma is the most common type (85% of cases) of pancreatic cancer, arising in exocrine glands, while pancreatic neuroendocrine tumor (PanNET) is less common (less than 5%) and occurs in the endocrine tissue of the pancreas [4]. Intraductal Papillary Mucinous Neoplasm (IPMN) is a cystic neoplasm identified as a precursor lesion for pancreatic cancer. Depending on the location of IPMN in the ductal system, it has a considerable risk of malignancy [5,6,7]. In Brazil, pancreatic cancer accounts for 2% of all cancers and for 4% of death, and in 2022 the INCA (Cancer National Institute) estimated 10,980 new cases (https://www.gov.br/inca/pt-br/assuntos/cancer/tipos/pancreas, accessed on 10 July 2023).

Pancreatic cancer patients have the lowest survival rate of all major organ cancers, and this rate is closely related to the tumor stage [8]. A recent report by Bengtsson [8] looked up pancreatic cancer long-term survival (≥5 years) using the National Cancer Institute’s Surveillance, Epidemiology, and End Results (SEER) database between 1975 and 2011, where 5-year survival for all stages rose from 0.9% in 1975 to 4.2% in 2011. Recent data from 2021 from the American Cancer Society showed that for all stages combined, the 5-year relative survival rate is 10% [9]. Pancreatic cancer usually undergoes late diagnosis in advanced tumor stages due to a wide range of nonspecific symptoms. The more common clinical symptoms are weight loss, jaundice, abdominal pain, anorexia, and dark urine [10]. Regarding image-based tests, abdominal ultrasonography, triphasic pancreatic-protocol CT (arterial, late, and venous phases) cross-sectional imaging, and magnetic resonance imaging (MRI) are useful tools. If a pancreatic mass is identified, subsequent endoscopic ultrasonography and fine-needle aspiration are used for cytological diagnosis [10,11,12,13].

A combination of complete resection and systemic multi-agent chemotherapy is the only hope to cure pancreatic cancer. Surgical outcomes have improved due to increased surgical expertise, and more patients are currently considered for resection. However, the overall recurrence rate remains at 70–80%, commonly occurring as distant metastases to the liver, which drove current practice toward neoadjuvant chemotherapy before surgery, regardless of resectability. Current NCCN guidelines recommend Folfirinox or Gemcitabine in combination with nab-paclitaxel as preferred agents. Adjuvant chemotherapy after curative intent surgery in PDAC is evolving with therapies with modified Folfirinox for 24 weeks and Gemcitabine with or without Capecitabine in cases of intolerability to mFolfirinox. Generally, locally advanced or metastatic PDAC patients are considered non-curative and managed with palliative chemotherapy with gemcitabine monotherapy or Folfirinox [14].

Plasma biomarkers for pancreatic cancer diagnosis and prognosis have been suggested, such as specific exosome and plasma microRNAs that are elevated in pancreatic cancer patients [15,16]. In addition, specific mutations in exosome-derived DNA, for example, *KRAS* and *TP53*, have been reported in pancreatic cancer patients [17,18]. This highlights the potential of exploiting exosome composition as a source for cancer biomarkers. In the case of protein biomarkers, CA-19-9 antigen may be helpful in symptomatic patients to confirm a diagnosis and predict prognosis and recurrence after resection (National Comprehensive Cancer Network 2014); however, besides presenting low sensitivity and specificity, it is not sufficient individual screening tool for asymptomatic patients [19,20,21]. Moreover, it cannot distinguish between cancer and chronic pancreatitis and possibly other disease states with chronic inflammation [22]. Therefore, the elucidation of new biomarkers is very important for an earlier diagnosis of the disease, as well as for a better prognosis. Efforts have been made to screen new protein biomarkers in cancer cell secretomes and plasma from pancreatic cancer patients [23,24]. Exosome-derived protein biomarkers are promising; however, few studies have been focused on them until now [25,26]. In this scenario, we propose a proteomics analysis of plasma-derived exosomes from patients with distinct types of pancreatic cancer and treatment, aiming to contribute to both biomarker screening and cancer cell signaling data.

## 2. Results

### 2.1. Epidemiological Data

Epidemiological data for gender, age, tobacco and alcohol usage, and diabetes for both groups are represented in Table 1, demonstrating that no significant difference was observed between these parameters.

### 2.2. Nanoparticle Tracking Analysis

The average size of exosomes obtained from healthy controls (114.13 ± 13.70 nm) was significantly lower than that of pancreatic cancer patients (117.3 ± 24.68 nm), (F-value = 3.246; *p*-value = 0.0351; df = 14). Regarding the average concentration of exosomes, a significant difference (t = 6.088; *p* ≤ 0.0001; df = 15.09) was observed between the healthy controls (8.83 × 10^11^ ± 3.5 × 10^11^ particles/mL) and pancreatic cancer patients (3.70 × 10^12^ ± 1.8 × 10^12^ particles/mL). These results were published by Marin et al., 2022 [16].

### 2.3. Proteome Profiling of Plasma Exosomes Derived from Pancreatic Cancer Patients

This study explored the proteomic landscape of plasma exosomes derived from IPMN (N = 7) and PDAC-S (N = 5) patients, as well as healthy controls (N = 10). Additionally, the proteome of exosomes from PDAC patients treated with neoadjuvant Folfirinox therapy (PDAC-F N = 4), and Gemcitabine-based adjuvant therapy (PDAC-G; N = 7) were analyzed, totalizing 33 patient samples. A high-throughput label-free mass-spectrometry approach was employed to identify and quantify the proteome of the exosomes. Subsequently, statistical and bioinformatics analysis were applied to explore the changes in protein expression across the different sample groups to assess the biological significance of these exosome proteins in the context of tumorigenesis. Figure 1 provides an overview of the study workflow.

Initially, 319 different proteins were identified in the exosome samples, with a false discovery rate (FDR) of 1% (on average, approximately 173 proteins per sample). However, for a more reliable and representative proteome, only the 135 proteins expressed in at least 70% of the samples in each group were retained for further analysis.

In the study’s initial sections, we explored the differential proteome of IPMN and PDAC-S exosomes by comparing their proteomic content with that of healthy control exosomes (IPMN vs. CT; PDAC-S vs. CT). Moreover, we analyzed the exosome proteomes between patients with these distinct pancreatic diseases (PDAC-S vs. IPMN). Finally, we investigated the differential proteome of the exosomes from PDAC patients that underwent only surgery and therapy-treated PDAC patients (PDAC-S vs. PDAC-F; PDAC-S vs. PDAC-G). The identified proteins are presented using their corresponding coding gene symbols.

### 2.4. Characterization of the Differential Proteome from Plasma Exosomes Derived from IPMN and PDAC-S Patients

In the IPMN vs. CT, PDAC-S vs. CT, and PDAC-S vs. IPMN comparisons (*p*-value < 0.05, logFC ± 0.58), we identified 33, 34 and seven differentially expressed proteins (DEPs), respectively (Figure 2A). KLKB1, LBP, CFB, and SERPINA1 were the most over-expressed proteins in the IPMN exosomes compared to the control group, and C5, APOD, C3, and C1QA were the most hypo-expressed DEPs. In the PDAC-S proteome, CPN1, IGHV2-26, ITIH3, and CLU were the most over-expressed, and C4BPB, APOB, CFH, and C1QB were the most hypo-expressed DEPs regarding control samples (Appendix A).

In the comparisons of IPMN vs. CT and PDAC-S vs. CT, a total of 29.17% of the proteins were found to be uniquely differentially expressed in the IPMN comparison. The proteins in this group included LBP, SERPINA4, F2, and ITIH4. Similarly, 31.25% of the proteins were exclusively differentially expressed in PDAC-S exosomes, including proteins such as APOB, C8A, IGKV2D-28, and IGLL5 (Figure 2B).

The other 39.58% of the identified DEPs were commonly identified in both comparisons. This group of DEPs includes the over-expressed AGT, SERPINA1, SERPING1, ITIH3, and KNG1, and the hypo-expressed C3. Only two proteins, IGHV5-51, and A2M were exclusively identified as DEPs in the PDAC-S vs. IPMN comparison, both were over-expressed in PDAC-S. Principal component analysis (PCA) revealed that the differential expression profile of identified proteins in IPMN (Figure 2C) and PDAC-S (Figure 2D) can adequately distinguish IPMN and PDAC-S patients from healthy controls.

### 2.5. Protein–Protein Interaction Networks and Enrichment Analysis of the IPMN and PDAC-S Exosome DEPs

To investigate the functions of the identified DEPs in IPMN and PDAC-S samples, as well as the biological processes and pathways that can be impacted by their dysregulated expression, we employed the STRING database (v. 11.5) to predict the main interactions involving each set of DEPs. We also performed enrichment analysis based on the Reactome pathways, GO-MF, and GO-BP collections from MSigDB v. 2023.1 using the IPMN and PDAC-S DEPs separately (FDR < 0.05). The interaction networks exhibited significantly more interactions than expected, indicating that the proteins are biologically connected, at least partially, as a group (Figure 3). This observation suggests their potential involvement in coordinated biological processes and pathways.

The enrichment analysis reveals distinct biological alterations that were uniquely enriched in the IPMN DEPs, PDAC-S DEPs, and some alterations that were common to both diseases (Figure 4A–D; Appendix A).

In the IPMN vs. CT set, the DEPs were especially involved in inflammatory response (FDR = 8.95 × 10^−14^) and acute inflammatory response (2.95 × 10^−13^), involving DEPs such as AGT, ITIH4, F2, and HP; activation of C3 and C5 (FDR = 3.94 × 10^−6^), with the enriched proteins C3, C5, and CFB; and neutrophil degranulation (FDR = 2.84 × 10^−3^), involving A1BG, SERPINA1, and SERPINA3 proteins.

In the PDAC-S vs. CT comparison, the DEPs were involved in the chylomicron assembly and remodeling alterations (FDR = 6.61 × 10^−6^), involving the apolipoproteins APOA1, APOA2, and APOB; FCGR activation (FDR = 2.61 × 10^−5^) and CD22 mediated regulation (FDR = 1.68 × 10^−5^), enriched in immunoglobulins IGKV1-16, IGKV2D-28, IGKV3-15, and IGLV1-40; and wound-healing processes (FDR = 2.44 × 10^−8^), in which the proteins KNG1, SERPINA1, SERPING1, and VTN were identified.

Furthermore, some alterations were enriched in both sets of DEPs (IPMN and PDAC-S), including the complement cascade (FDR = 4.77 × 10^−20^ and 9.72 × 10^−30^, respectively), hemostasis (FDR = 1.11 × 10^−17^ and 2.3 × 10^−19^, respectively), innate immune system (FDR = 3.49 × 10^−16^ and 1.93 × 10^−20^, respectively), and intrinsic pathway of fibrin clot formation (FDR = 3.73 × 10^−12^ and 1.47 × 10^−9^, respectively). Altogether, C3, F2, SERPING1, CLU, SERPINA1, VTN, and A1BG were identified in these processes.

Additionally, we also explored the hallmark and cancer module collections of MSigDB v. 2023.1 (Appendix A). Some hallmarks were commonly enriched for the IPMN and PDAC-S DEPs, such as complement (FDR = 1.41 × 10^−20^ and 1.24 × 10^−16^, respectively), coagulation (FDR = 8.39 × 10^−17^ and 2.17 × 10^−20^, respectively), KRAS signaling (FDR = 6.31 × 10^−3^ and 5.52 × 10^−3^, respectively), and xenobiotic metabolism hallmarks (FDR = 6.31 × 10^−3^ and 5.52 × 10^−3^, respectively), while the interferon-gamma response (FDR = 5.52 × 10^−3^) was the only hallmark enriched for the PDAC-S DEPs. In general, the enrichment analysis based on the cancer modulation corroborates the pathways and GO results and reveals that many of the proteins identified in this study have already been associated with various types of cancer and tumorigenesis processes.

### 2.6. Identification of Oncoproteins and Tumor-Suppressor Proteins among the IPMN and PDAC-S Exosome-Derived DEPs

The sets of DEPs identified in this study were analyzed using the NCG (v. 7.1) database to explore their association with cancer-related genes. Among all the DEPs, we identified seven potential or canonical oncogenes and tumor suppressor genes. Table 2 summarizes the gene identification, expression pattern, cancer driver classification, and enriched biological pathways associated with each DEP.

### 2.7. Influence of Treatments in Protein Expression of PDAC Patients

The comparative analysis of exosome proteomes also aimed to identify proteins whose expression was altered after treatment in patients with PDAC, highlighting potential biomarkers of therapy response and follow-up. This section of the study included those patients who underwent surgery only (PDAC-S) as well as those who received Folfirinox neoadjuvant therapy (PDAC-F) or Gemcitabine adjuvant therapy (PDAC-G). In total, eleven DEPs were identified in the comparisons between these groups, including five immunoglobulins, AGT, C8A, ORM1, F2, SERPINF1, and FCN3 (Figure 5A). Further, only the protein ORM1 was differentially expressed between PDAC-F and PDAC g samples (Appendix A).

Interestingly, AGT, IGKV2D-28, and C8A presented notable expression patterns (Figure 5B). AGT exhibited overexpression in PDAC-S tumors compared to healthy controls; however, its expression levels decreased when comparing Folfirinox and Gemcitabine therapies to controls. Similar patterns were observed for C8A, particularly in the case of Folfirinox adjuvant therapy. Similarly, IGKV2D-28, which was downregulated in PDAC-S samples compared to healthy controls, exhibited increased expression after Folfirinox and Gemcitabine therapy when compared to PDAC-S, reaching expression levels similar to those of healthy controls.

The DEPs set related to the therapies were also identified in biological pathways and processes according to the MSigDB v. 2023.1 database (Table 3). In this analysis, the enriched biological processes were mainly related to immune response and complement cascade, including signaling pathways such as G-alpha signaling events (FDR = 3.65 × 10^−2^), FceRI mediated NF-κB activation (FDR = 1.73 × 10^−4^), and FceRI mediated MAPK activation (FDR = 6.93 × 10^−5^).

Finally, the PDAC-F and PDAC-G samples were stratified based on treatment duration using a six-month cutoff, considering short (<6) and long (>6) treatment periods, and compared to PDAC-S (*p*-value < 0.05, logFC ± 0.58). The results of this analysis are described in Appendix A. Some DEPs, such as SERPINA3, HPX, GSN, LBP, C1QB, and C1QC, exhibited altered expression specifically in certain treatment groups and duration times.

## 3. Discussion

In this study, we analyzed plasma-derived exosomes from pancreatic cancer patients in a miscellaneous cohort. For the statistical and enrichment analysis many types of comparisons were performed showing that the major differences in protein expression are in PDAC and IPMN. PDAC tumor is the most studied, frequent, and aggressive type of pancreatic cancer [27] while IPMN is a cystic lesion that, depending on the location in the duct system, can be considered at high-level risk to become malignant [28].

In IPMN samples, the most notable alterations observed in the proteomics profile compared to healthy controls were the over-expression of KLKB1, LBP, SERPINA1, and CFB, and the downregulation of C3, C5, APOD, and C1QA. The plasma kallikrein (KLKB1) is recurrently suggested as a biomarker for different types of carcinoma [29], but has also been previously detected on serum samples from IPMN patients [30], and benign pancreatic diseases [31], while alpha-1-antitrypsin (SERPINA1) over-expression was previously detected in plasma and suggested as a biomarker for pancreatic cancer, influencing inflammatory response, blood clotting, and immunity [32,33] Otherwise, the decreased expression of the human plasmatic apolipoprotein D (APOD) has been related to poor survival and worse prognosis in several cancer types [34].

The crosstalk between inflammation and immune alterations is closely associated with tumor predisposition, driving malignant initiation, conversion, and growth, as well as taking part in advanced steps such as invasion and metastasis. Interestingly, we show here that inflammatory and immune system-related alterations, such as activation of C3 and C5, acute inflammatory response, neutrophil degranulation, and complement cascade, stand out as altered processes in IPMN due to exosome protein dysregulations. CFB, C3, and C5, components of the complement system, along with LBP, APOD, and SERPINA1, were some of the DEPs associated with these processes on IPMN.

CFB, C3, and C5 are commonly associated with changes in the tumor microenvironment. These proteins are linked to abundant immune infiltrates, immune responses, and senescence, and serve as prognostic markers for various types of cancer [35,36,37], including pancreatic diseases [38,39]. In pancreatic cancer, the accumulation of CFB is associated with immune evasion, making it a potential target for immunotherapy [40]. On the other hand, C3 and C5 have been suggested as targets for immunotherapy and as follow-up markers in pan-cancer scenarios [38]. Expression changes in CFB, C3, and C5 in pre-neoplastic lesions such as IPMN suggest that immune alterations precede the establishment of pancreatic neoplasia.

LBP was found to be up-regulated in IPMN compared to healthy controls. LBP is an LPS (lipopolysaccharide)-binding protein that has a positive correlation with PD-L1 (programmed death ligand-1). Yin and collaborators [41] showed an important pathway in pancreatic cancer in which LBP acts. LPS is involved in immune maturation and activation, and recent studies have described these molecules, as well as bacterial DNA in tumor tissue [41,42,43]. Importantly, in the last update of Hallmarks of Cancer [44], a new Hallmark named “Polymorphic microbiomes” was included. There is a communication/exchange of metabolites and/or signal molecules between microorganisms and host cells that can positively or negatively affect the tumor microenvironment [44]. LBP could increase tumor-infiltrated lymphocytes, suggesting that LPS has potential as an immunological adjuvant in pancreatic cancer [41].

Altogether, KLKB1, SERPINA1, CFB, C3, and C5 represent DEPs on IPMN that could be further investigated in the context of pancreatic tumorigenesis. These molecular alterations and their associated biological pathways and functions can help to understand the pathological processes that contribute to cancer progression. Moreover, these DEPs could be potential biomarkers for the early detection of IPMN through liquid biopsy methods.

PDAC comprises a highly aggressive and lethal form of pancreatic cancer with a notable ability to evade the immune system, creating a tumor microenvironment that promotes immune tolerance and resistance to immune-mediated destruction. This immune evasion, along with other factors, contributes to the aggressive nature of PDAC and its resistance to standard cancer therapies. Our results showed many DEPs related to the immune system and correlated processes. Recently, Huang and collaborators [45] have compiled the recent advances in PDAC proteomic studies and explored how they can contribute to clinical development. They showed results using plasma, tissue, and exosomes proposing a panel of proteins that together have the potential as diagnosis and prognosis biomarkers.

Our proteomic analysis revealed significant differential expression of several proteins in PDAC-S samples compared to healthy controls. Notably, CPN1, IGHV2-26, ITIH3, and CLU were found to be overexpressed, while C4BPB, APOB, CFH, and C1QB showed downregulation in PDAC-S samples.

Carboxypeptidases (CPs) form an extensive group of zinc metallopeptidases responsible for various physiological roles by removing C-terminal basic residues from proteins and peptides [46]. Carboxypeptidase N (CPN) plays a crucial role in regulating vasoactive peptide hormones, growth factors, and cytokines, which are typically secreted by cells in the tumor microenvironment [47]. CPN acts in the crosstalk between coagulation, thrombosis, inflammation, and innate immunity [48], which were among the most enriched pathways found for our DEPs.

CPs contribute to the pathogenesis of multiple cancer types [47]. In breast cancer, high levels of CPN were described, which contribute to the cleavage of specific polypeptide fragments within the tumor microenvironment [49]. In pancreatic tumors, other CPs have been reported, such as the carboxypeptidase E promoting proliferation [50] and regulating the transcriptional and epigenetic profiles. The carboxypeptidase A1 is a highly sensitive marker for pancreatic acinar cell carcinoma [51]. In this study, we identified increased levels of CPN1 in PDAC-S as compared to healthy controls, thereby expanding the CP repertoire in pancreatic tumorigenesis.

ITIH3, another overexpressed DEP in the PDAC-S samples, also acts in the tumor microenvironment contributing to tumor progression. The inter-alpha-trypsin inhibitors (ITI) constitute a family of plasma protease inhibitors that contribute to the stability of the extracellular matrix through covalent binding to hyaluronan [52]. The growth of tumors is associated with an increase in the size of the epithelial hyaluronic complex [53], emphasizing the important role of ITI members in tumorigenesis. ITIH plays an important role both in inflammation and in carcinogenesis [52]. Our exosome proteomics analysis revealed high levels of ITHI3 in PDAC-S patients compared to healthy controls, and its differential expression according to the time of Folfirinox treatment. Consistently, ITIH3, which has significant involvement in extracellular matrix remodeling during tumor progression, along with APOA1, APOE, APOL1, and CA19-9, constitutes a biomarker panel for pancreatic cancer [54].

Clusterin (CLU) is a highly evolutionary conserved glycoprotein that controls crucial physiological processes and several cancer-associated events, including cancer cell proliferation, stemness, survival, metastasis, epithelial-mesenchymal transition (EMT), therapy resistance, and inhibition of programmed cell death, which in turn facilitates cancer growth and recurrence [55,56]. This regulation occurs through diverse signaling pathways and contributes to the progression of various cancers, such as prostate, breast, lung, liver, colon, bladder, and pancreatic cancer [55]. In pancreatic cancer cell lines, CLU was shown to inhibit tumor proliferation, 3D spheroid growth, invasiveness, EMT, and reduce sensitivity to gemcitabine therapy [57]. CLU knockout represses proliferation in pancreatic cancer by inducing cellular senescence [58]. The controversial data on the regulation of CLU levels might be related to the fact that CLU is not uniformly expressed in pancreatic cancer and may even have distinct and conflicting roles in tumorigenesis, depending on the origin of the tumor [59]. Our study revealed a higher average expression of CLU in PDAC-S (1.5-fold increase) and IPMN (1.07-fold increase) samples compared to healthy controls.

Our findings revealed the downregulation of C4BP in PDAC-S samples compared to healthy controls. C4b-binding protein (C4BP) is a prominent regulator of the complement system [60], acting as the primary fluid phase inhibitor of the classical and lectin pathways of complement activation [60,61]. In pancreatic tumors, a comprehensive proteomic study identified the C4b-binding protein α-chain (C4BPA) as a novel serum biomarker, showing promise for early stage PDAC detection and differentiation from other gastroenterological cancers [62]. It has been reported that stimulation with murine C4BPA peptide increased the number of CD8+ tumor-infiltrating lymphocytes surrounding PDAC tumors in vivo and high stromal C4BPA was associated with favorable PDAC prognosis [63]. In this regard, C4BP downregulation in PDAC-S would be indicative of poor lymphocytic infiltration in our samples.

Another component of the complement cascade that was downregulated in PDAC-S is the complement C1q subcomponent subunit (C1QB). The complement cascade, as an integral part of innate immunity, not only acts as a critical mediator in the innate defense against pathogens but also plays a regulatory and anti-inflammatory role in the clearance of immune complexes and dying cells from damaged tissues [60,64,65]. Its role in cancer and tumor microenvironment has been explored. C1QB can mediate growth factor-induced cancer cell chemotaxis and distant metastasis, including to the liver, which is a significant event in the progression of pancreatic cancer and is associated with an extremely poor prognosis [66]. Moreover, C1QB levels in pancreatic cancer may serve as a predictor of disease [67] given its role in the regulation of IGF-1/IGF-1R signaling, which plays a role in the cell spreading, and in the induction of hepatic metastasis [66]. In this study, C1QB was downregulated in PDAC-S compared to healthy controls, while it was overexpressed in PDAC g (>6) compared to the PDAC-S, which may be explained by the metastatic and more advanced stage profile of the patients treated with gemcitabine.

The enrichment analysis of our differentially expressed proteins (DEPs) indicates the potential relevance of innate immune and complement systems, hemostasis, the coagulation cascade, as well as processes associated with IGF regulation, platelet calcium cytosolic levels, and G protein signaling. These findings suggest the involvement of these biological pathways and functions in pancreatic tumorigenesis. In PDAC, the significance of innate immunity has been widely explored due to the presence of intracellular functions for innate immunity proteins within tumor cells [68]. Most of our interesting DEPs that were identified are part of immune system-related processes. This highlights the dynamic nature of the tumor microenvironment and emphasizes the importance of further investigating the interplay between the innate immune system and the tumor development and progression aiming to contribute to novel therapeutic approaches.

The complement system plays a role in the immune surveillance of pathogens and tumor cells. Proteomic studies revealed that extracellular vesicles (EVs) released by metastatic hepatocellular carcinoma (HCC) cells contained a substantial presence of complement proteins [69]. In accordance, our data show several components, molecular functions, and biological pathways related to the complement cascade. Emerging evidence highlights the importance of complement proteins in tumor formation and cancer metastasis, with complement proteins being related to pro-tumoral and anti-tumoral roles across different cancer types [69]. These findings provide a potential explanation for the complement repertoire observed in our IPMN and PDAC compared to healthy controls. However, when we directly compare IPMN and PDAC-S proteomes, only IGHV5-51 and A2M are exclusive DEPs, and complement proteins are not found in this comparison, indicating an early dysregulation of the complement system in pancreatic cancer.

The hallmarks enriched for our exosome proteomes (IPMN vs. CT and PDAC-S vs. CT) support the main biological pathways and functions discussed earlier, particularly the complement system and related DEPs, such as C3, C4BPB, CFB, CFH, CLU, KLKB1, SERPINA1, and SERPING1. Most of them were also observed in the coagulation hallmark. Interestingly, CFB and SERPINA3, which exhibited higher expression in IPMN and PDAC-S exosomes, have been associated with KRAS signaling [70,71]. Oncogenic mutant KRAS is implicated as the primary driver in the initiation of PDAC, with activating mutations of KRAS commonly observed in PDAC. Thus, our findings suggest a role of KRAS signaling and the relevance of CFB and SERPINA3 in pancreatic tumorigenesis.

Li and collaborators [72] performed a high-throughput study using proteomic approaches that showed that 4 proteins (ITIH3, APOA1, APOE, and APOL1) combined with CA19-9 had improved sensitivity and specificity to PDAC diagnosis. Another proteomic study in diabetic patients with PDAC validated a panel of potential biomarkers, in combination with CA19-9. This panel consisted of apolipoprotein A-IV (APOA4), monocyte differentiation antigen CD14 (CD14), tetranectin (CLEC3B), gelsolin (GSN), histidine-rich glycoprotein (HRG), inter-alpha-trypsin inhibitor heavy chain H3 (ITIH3), plasma kallikrein (KLKB1), leucine-rich alpha-2-glycoprotein (LRG1), pigment epithelium-derived factor (SERPINF1), plasma protease C1 inhibitor (SERPING1), and metalloproteinase inhibitor 1 (TIMP1), and demonstrated an area under the curve (AUC) of 0.85 and an increased accuracy compared to CA19-9 alone [73]. Tonack and collaborators [74] have also looked for potential plasma biomarkers for PDAC and found some candidates, but they are associated with jaundice, showing the highest sensitivity of ITIH3, C5, A1BG, PIGR, and Reg3A in this condition. Despite the use of different cohorts and workflows, from the 18 proteins reported in the studies above, 9 of them were DEPs in the present study: A1BG, C5, SERPING1, SERPINF1, KLKB1, HRG, APOL1, APOA1, and ITIH3. This reinforces their participation in the cancer mechanism and potential as biomarkers. ITIH3 was found to be over-expressed in IPMN and in PDAC-S conditions compared to the control. ITIH3 belongs to the inter-α trypsin inhibitor (ITI) family of serine protease inhibitors, which are involved in the stabilization of extracellular matrix by covalently binding to hyaluronic acid [75], and has also been described to be overexpressed in plasma from gastric cancer patients [76].

The family of apolipoproteins is associated with a variety of cancer types. APOA1 was overexpressed in PDAC-S compared to control, and APOB was downregulated in the plasma of PDAC-S patients. Apolipoprotein A (APOA) seems to be a good indicator of several cancers, such as colon, hepatocellular, and pancreatic cancer; meanwhile, apolipoprotein E (APOE) may have polymorphisms that affect tumor susceptibility. APOC, APOB, and APOD are involved in tumor progression [77]. ApoA1 was described in a biomarker panel together with other five proteins (CA125, CA19-9, CEA, ApoA2, and TTR), with parameters of the area under the curve, specificity, and sensitivity of 0.992, 95%, and 96%, respectively [78]. The axis TRIM15-APOA1-LDLR may be a target for treatment given its involvement with PDAC metastasis; TRIM15 interacts with APOA1, promoting APOA1 polyubiquitination, and consequently its degradation. This results in enhanced lipid anabolism, and accumulation of lipid droplets in pancreatic cancer cells, a common metabolism dysregulation in PDAC [79].

APOB is downregulated in CRC (colorectal carcinoma), and its silencing in HCC (hepatocellular carcinoma) resulted in increased cell proliferation rates [80,81], indicating an anti-growth property of APOB. In HCC, low expression of APOB was related to increased expression of metastatic and oncogenic regulators (FOXM1, MTIF, HGF, CD44, and ERB2) and to the downregulation of tumor suppressors, such as PTEN and TP53. Due to this, the inactivation of APOB in HCC indicates a poor prognosis [81]. In HCC, APOB can be altered by somatic mutations or hypermethylation, resulting in the diverting of energy to cancer metabolic pathways [82]. Germline mutations and polymorphisms in APOB are involved in metabolic disorders, resulting in abnormal lipid metabolism, which is involved in the promotion of pancreatic tumorigenesis [83,84]. An exome-wide analysis in Chinese people identified 3 low-frequency missense variants associated with an increased risk of PDAC [85], one of those was rs183117027 in APOB. Ren and collaborators [34] compiled information about the members of the apolipoprotein family and its involvement in cancer. In pancreatic cancer, APOA2, APOC1, APOC2, and APOJ have been described, while APOB was found in HCC, bladder, and breast cancer. We have found APOB downregulated in PDAC-S condition as compared to controls, suggesting its involvement also in pancreatic cancer development. Considering the evidence that a lipid metabolism disorder is involved in pancreatic cancer and that apolipoproteins are involved in this, there is a clinical trial (https://clinicaltrials.gov/ct2/show/NCT04862260, accessed on 10 July 2023) in course. This study aims to verify if the addition of a cholesterol shortage on top of Folfirinox treatment in newly diagnosed patients (with local or metastatic PDAC), can lead to a better progression in tumor treatment response. It is expected that a drug-induced cholesterol shortage will slow down or stop the progression of pancreatic adenocarcinomas while increasing the response to chemotherapy (https://clinicaltrials.gov/ct2/show/NCT04862260, accessed on 10 July 2023).

Another protein family largely associated with cancer is the complement system proteins. In the context of chemotherapy response, the complement system, together with the tumor microenvironment (TME) plays pivotal roles in cancer progression, and therapy failure [86]. The complement system links innate to adaptive immunity and is essential for the removal of apoptotic cells or foreign substances, by triggering a cascade of enzymatic events that stimulates phagocytosis by immune cells [87]. The complement system has several proteins, C3 being the most abundant, and many studies have shown its role in cancer pathways [35,38]. A multi-omics analysis has found an association between C3/C5/C3AR1/C5AR1 and tumor immune evasion and therapy response in several types of cancer [38]. Nsingwane and collaborators [88], found that C3 plasma levels were decreased in locally advanced and metastatic disease compared to resectable cancer, suggesting a loss of innate immune response. Proteomic analysis of the secretome of tumorigenic cell lines (AsPC1, MIA PaCa-2, and PANC1) showed inhibition of the complement system compared to the HPDE6 cell line. This can allow these cells to survive the attack of secreted components of the complement system [89]. In the current study, C3 was found to be downregulated in IPMN and PDAC-S when compared to controls, suggesting that the complement system is also involved in IPMN development.

SERPINA1, SERPINA3, SERPINC1, SERPINAC1, and SERPINAG1 were upregulated in PDAC-S and IPMN as compared to healthy controls, but not in the direct comparison between IPMN and PDAC-S. The proteins coded by these genes are involved in the complement and coagulation cascade (KEGG Pathways—map04610: complement and coagulation cascade). Wu and collaborators [32], looked for glycobiomarkers of pancreatic cancer by iTRAQ quantitative proteomics and found elevated levels of 22 glycopeptides in the plasma of pancreatic cancer patients. Among these glycopeptides was the fucosylated SERPINA1 (fuco-SERPINA1), which was further validated, being associated with higher TNM stages and poor prognosis, suggesting that this molecule is a prognosis biomarker.

SERPINA3 is an acute-phase protein, so its expression is not a cancer-specific protein and increases in response to inflammation. It acts as an inhibitor of several serine proteases, including pancreatic chymotrypsin and cationic elastase. The expression of SERPINA3 has been previously reported to be increased in tumoral tissues and sera from patients with PDAC, with evidence for correlation with poor survival [90].

SERPINA4 was up-regulated only in IPMN, compared to controls. Some studies showed that SERPINA4 [91,92] has an important effect on the inhibition of tumor growth and angiogenesis. Sun and collaborators [93] investigated the expression of SERPINA4 and its clinical significance in colorectal cancer (CRC) and concluded that the decrease in SERPINA4 expression is associated with invasion depth, nodal involvement, distant metastasis, tumor stage, and differentiation. In addition, they suggested that SERPINA4 can be used as a good prognostic indicator and has potential as a therapeutic target for CRC. Zhu and collaborators [94], also found increased expression of SERPINA4 in PDAC patients, suggesting its use as a biomarker.

Considering treatments, AGT was overexpressed in PDAC-S compared to both PDAC g and PDAC-F, while C8A was overexpressed only in PDAC-S compared to PDAC-F, along with ORM1 and SERPINF1. These DEPs may be involved in the drug response but also may reflect the advanced stage of the patient’s receiving chemotherapy. Among hypo-expressed proteins, a member of the immunoglobulin variable chains appears both in PDAC-S versus PDAC-F and PDAC-G.

Here, we identified DEP from plasma exosome-derived proteomic profiles of IPMN and pancreatic cancer patients, subdivided into two antagonistic tumor phenotypes, comparing those to healthy control samples. Despite most cases of IPMN being asymptomatic, the molecular changes involved in the IPMN pathology can lead to physiological modifications, creating a favorable environment for the development of PDAC. The early identification of IPMN lesions and comprehension of their associated biological characteristics can be useful to guide patient management and follow-up. Hence, the proteomic alterations observed in the exosomes from IPMN patients can represent potential biomarkers for the detection of IPMN and provide insights into how IPMN influences the tumor microenvironment and neoplastic progression. Moreover, the DEPs identified in PDAC described here could represent biomarkers for diagnostic and prognostic evaluation, which, in association with CA19-9 could potentially improve pancreatic cancer screening and early diagnosis. This report corroborates previous findings and provides additional scientific data on the presence, role, and potential application of exosome proteins to the basic and clinical knowledge of pancreatic cancer. The weakness of this report relies on the reduced number of samples within each pancreatic cancer group, which makes sample heterogeneity a limitation of the findings. However, this report provides data to improve follow-up with specific targets that could be useful as plasma biomarkers for pancreatic cancer. An increased number of samples associated with clinical and treatment data, and especially patient follow-up, should be used to validate such proteins as exosome-derived plasmatic biomarkers in pancreatic cancer.

As future perspectives and goals, our group will focus on the validation of the proteins described here in a larger cohort of pancreatic cancer patients. Within this, we highlight a cohort of sequential samples from the same patients in order to perform a temporal evaluation of these proteins. Another future perspective relies on the validation of this finding in an in vivo approach by looking for an exosomal protein biomarker in mouse pancreatic cancer models.

## 4. Methods

### 4.1. Ethical Statement and Samples

This work was conducted after approval from the Ethics Committee of Fiocruz, Instituto do Cancer do Estado de São Paulo (ICESP) and Hospital do Trabalhador (CAAE 89520218.7.0000.5248, 77979417.8.0000.5248, and 77979417.8.3001.5225). All sampling and experiments were performed following relevant guidelines, Brazilian regulations, and ethical principles for human research in the Declaration of Helsinki. The project was described to all participants, and a written informed consent and epidemiological questionnaire were obtained from all participants enrolled in the study. A total of 23 pancreatic cancer patients (16 PDAC and 7 IPMN) were recruited from April 2018 to September 2019 with the inclusion criteria as disease confirmation by histopathology and/or surgery at the moment of sample collection. The PDAC patients were subdivided into three groups according to treatment undergone in sample collection, as follows: PDAC-S (non-treatment, N = 5), PDAC-F (Folfirinox treated, N = 4), and PDAC g (Gemcitabine treated, N = 7). All pancreatic cancer patients were provided from the Academic Biobank for Research on Cancer at the University of Sao Paulo (USP) (Biobank-USP), located at the Center for Translational Research in Oncology, São Paulo State Cancer Institute (Centro de Investigação Translacional em Oncologia, Instituto do Cancer do Estado de São Paulo-ICESP), São Paulo, Brazil. The Biobank–USP protocol was approved by the Local Ethics Committee (CEP no. 031/12 and the National Ethics Committee (CONEP no. 023/2014). As a control group, 10 non-cancer participants were recruited from Hospital do Trabalhador, Curitiba PR, Brazil, with the inclusion criteria of non-personal history of any kind of cancer. Demographic and epidemiological data were collected in both groups, while clinical data were collected for pancreatic cancer patients. For each patient, 4 mL of peripheral blood was collected in EDTA tubes, which were immediately centrifuged at 3000× *g* for 10 min, and the plasma was separated and stored at −80 °C.

### 4.2. Nanoparticle Isolation and Exosome Tracking Analysis

Nanoparticle isolation was performed with the commercial kit—miRCURY^®^ Exosome Serum/Plasma Kit (Qiagen cat number ID: 76603), following the manufacturer’s instructions to an initial volume of 0.6 mL plasma. Exosome Tracking Analysis using Nanosight LM-10 (Malvern Panalytical, Malvern, UK) was used to determine the concentration and size of exosomes isolated from the blood plasma of pancreatic cancer patients and controls. Exosome samples were diluted in PBS before injection for cell range from 40 to 100 by the reading frame. The videos were set to three runs each of 60 s, the detection threshold was defined as 4, and the camera level was 12. NTA statistical analysis was performed with Minitab Statistical Software 17.0, where samples were submitted to one-way variance analysis (ANOVA), and values were compared with Tukey’s test with a 5% probability level.

### 4.3. Sample Preparation for NanoLC-MS/MS

After the isolation of exosomes from plasma, the protein content of 33 samples was quantified by tryptophan fluorescence [95] then 20 μg of each sample was separated by SDS-PAGE 13% (*v*/*v*), followed by a standardized In-Gel digestion protocol adapted from [96]. Briefly, each lane was cut into small cubes and the proteins were reduced with 10 mM dithiothreitol and alkylated with 55 mM iodoacetamide before digestion with 12.5 ng/μL trypsin sequencing grade (Promega) in 50 mM ammonium bicarbonate (ABC) for 16–18 h at 37 °C. The following day peptides were eluted from the gel twice with 400 μL of 40% acetonitrile (ACN), 3% TFA, and twice with 400 μL of 100% ACN, then the collected fractions were dried using Speed Vac (Thermo Fisher Scientific, Waltham, MA, USA). The dried samples were resuspended in 100 μL of 0.1% formic acid and proceeded to desalination with an adapted protocol of stop-and-go extraction tips [97]. After desalination, the eluted samples were resuspended in 0.1% formic acid and the concentration of peptides was determined by absorbance (280 nm) in nanodrop (Thermo Fisher Scientific), then the peptide concentration of the samples was adjusted to 0.1 μg/μL before NanoLC-MS/MS analysis.

### 4.4. NanoLC-MS/MS Analysis

The digested samples (0.5 micrograms) were separated by online nanoscale capillary liquid chromatography and analyzed by nano electrospray tandem mass spectrometry (nanoLC MS/MS) in duplicate injection. The chromatography was performed on an Ultimate 3000 nanoLC (Thermo Fisher Scientific) followed by nanoelectrospray ionization, MS, and MS/MS on an Orbitrap Fusion Lumos (Thermo Fisher Scientific). The chromatographic conditions were as follows: mobile phase A 0.1% formic acid, mobile phase B 0.1% formic acid, and 95% acetonitrile. The flow of 250 nL/min, with a 90 min non-linear (Xcalibur type 6 curve) gradient from 5 to 40% B was conducted. The separation was performed on an in-house C18 packed emitter with 15 cm length, 75 μm Internal diameter, packed with 3.0 μm C18 particles (Dr. Maisch—ReproSil-Pur). MS and MS/MS scan parameters were as follows: MS1 acquisition in the Orbitrap analyzer with a resolution of 120,000 m/z window, from 300 to 1500 positive profile mode, with a maximum injection time of 50 ms. MS2 analysis was performed in data-dependent acquisition (DDA) mode of ions with 2–7 charges, 2 s per cycle where the most intense ions were subjected to high energy collisional dissociation (HCD) fragmentation at 30% normalized collision energy, followed by acquisition in the Orbitrap analyzer with a resolution of 15,000 in centroid mode. A dynamic exclusion list of 60 s was applied as well as the internal mass calibration for the MS1 scans. The nESI voltage was 2.3 kV and the ion transfer capillary temperature was 175 °C.

### 4.5. Data Analysis

The spectra identification was performed in MaxQuant version 1.6.17.0 [98] set as follows: specific search, trypsin as protease, carbamidomethylation of cysteine as fixed modification, oxidation on methionine, and acetylation on protein N-terminal as variable modifications, *Homo sapiens* reference database downloaded from Uniprot on 11 January 2021, containing 75,777 entries was used as a database, the reverse database used as a decoy for FDR estimation, 1% FDR for both PSM and protein assignment was accepted, the match between runs and LFQ intensity was enabled. Proteins were only considered for quantification if 2 or more unique peptides were identified for that group. The mass spectrometry proteomics data have been deposited to the ProteomeXchange Consortium via the PRIDE [99] partner repository with the dataset identifier PXD042543. All data processing and statistical analysis were conducted on the Perseus (v. 1.6.2.2) software, following the recommended pipeline suggested by the Perseus developers [98]. The normalized spectral label-free protein intensity (Label-free quantification; LFQ intensity) was used to determine protein expression, and only proteins identified in at least 70% of the samples of each group were maintained for the subsequent analysis. The LFQ values were logarithmized (log^2^) and normalized by Z-score, with missing values being replaced by values from the normal distribution (Width = 0.3; down-shift = 1.8). The normalization step was performed as recommended by Perseus developers [98].

The samples were grouped according to the following conditions: Controls (CT), IPMN, PDAC-S, PDAC-F, and PDAC-G. Student’s *t*-tests were applied (*p*-value < 0.05; logFC ± 0.58) to determine the differentially expressed proteins (DEPs) among the comparisons of IPMN vs. CT, PDAC-S vs. CT, IPMN vs. PDAC-S vs. IPMN, PDAC-S vs. PDAC-F, PDAC-S vs. PDAC-G, and PDAC-F vs. PDAC-G. The PDAC-F and PDAC g groups were also subdivided according to the duration of the therapy, using a six-month period as splitting criteria. Additional comparisons (PDAC-S vs. PDAC-F(<6) and PDAC-F(>6); PDAC-S vs. PDAC-G(<6) and PDAC-G(>6)) were performed to assess expression alterations related to therapy duration. Principal component analysis (PCAs) (Perseus) and heatmaps (complexheatmap R package) were used to investigate DEP’s expression patterns across comparisons. The interactive online tool [100] was accessed to visualize the distribution of DEPs across sample groups.

### 4.6. Protein–Protein Interaction Networks Construction and Functional Enrichment Analysis

The Molecular Signatures Database (MSigDB, v. 2023.1) [101] was accessed to perform a functional analysis of each set of DEPs separately. The enrichment analysis (FDR < 0.05) was performed according to the REACTOME, Hallmarks, and cancer modules collections, besides the molecular functions (GO-MF) and biological processes (GO-BP) datasets of the Gene Ontology consortium [102]. We further utilized the Network of Cancer Genes and Healthy Drivers (NCG, v. 7.1) database to explore whether our DEPs could be encoded by cancer driver genes categorized as canonical or putative oncogenes or tumor suppressor genes. The NCG classification is based on the prevalence of gene gain-of-functions or loss-of-functions alterations described in The Cancer Genome Atlas (TCGA) data [103].

Protein–protein interactions (PPI) were predicted using the Search Tool for the Retrieval of Interacting Genes/Proteins (STRING v. 10.5) [104] with the following parameters: data sources including “experiments”, “databases”, and “co-expression”, and a minimum interaction score of 0.70.

## Figures and Tables

**Figure 1 ijms-24-12604-f001:**
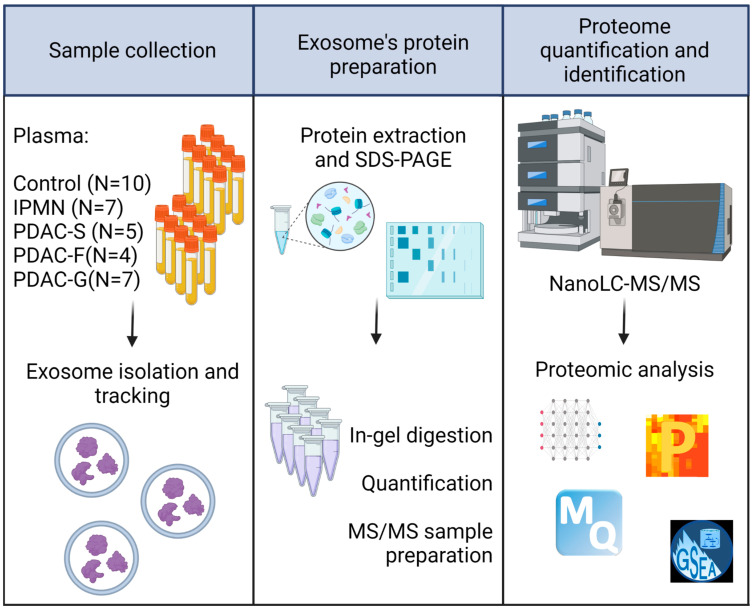
**Experimental workflow.** Schematic representation of the in-depth proteome profiling of plasma exosome proteins. IPMN = Intraductal Papillary Mucinous Neoplasm; PDAC-S = Pancreatic ductal adenocarcinoma—no treatment; PDAC-F = Pancreatic ductal adenocarcinoma—Folfirinox; PDAC-G = Pancreatic ductal adenocarcinoma—Gemcitabine.

**Figure 2 ijms-24-12604-f002:**
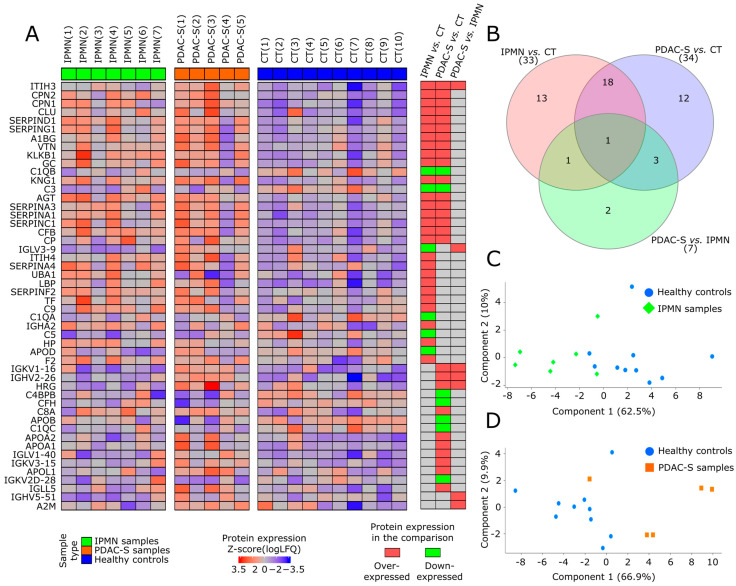
Differentially expressed exosome proteins were identified in distinct pancreatic diseases compared to the healthy controls. (**A**) Heatmap illustrating the expression pattern of the differentially expressed proteins (DEPs) identified in the IPMN vs. CT, PDAC-S vs. CT, and PDAC-S vs. IPMN comparisons (*p*-value < 0.05). (**B**) The Venn diagram illustrates the distribution of the DEPs across the comparisons. (**C**) The PCA analysis demonstrates the segregation of samples based on the DEPs of IPMN (green color) vs. CT comparison (blue color), and (**D**) PDAC-S (orange color) vs. CT (blue color) comparison. The heatmap (right) indicates the specific comparisons in which the proteins on each line exhibited differential expression. Red: Over-expressed protein in the comparison. Green: Hypo-expressed protein in the comparison. Grey: Protein non-differentially expressed in the comparison.

**Figure 3 ijms-24-12604-f003:**
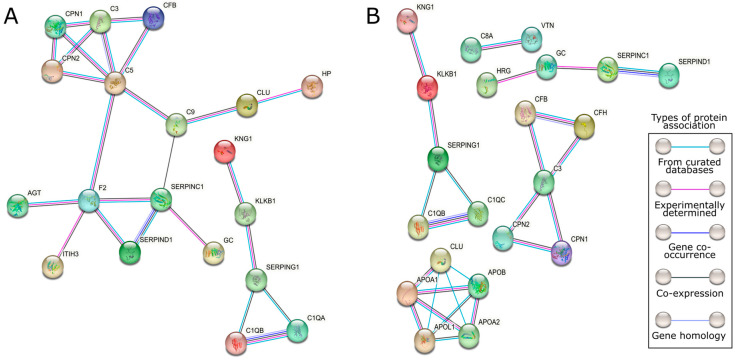
**Protein–protein interaction networks representing the association of the identified DEPs.** Protein associations were determined by the STRING (v. 11.5) database (PPI enrichment, *p*-value < 1.0 × 10^−16^). (**A**) PPI network constructed from the DEPs identified in the comparison of IPMN vs. CT samples. (**B**) PPI PPI network constructed from the DEPs identified in the comparison of PDAC-S vs. CT samples.

**Figure 4 ijms-24-12604-f004:**
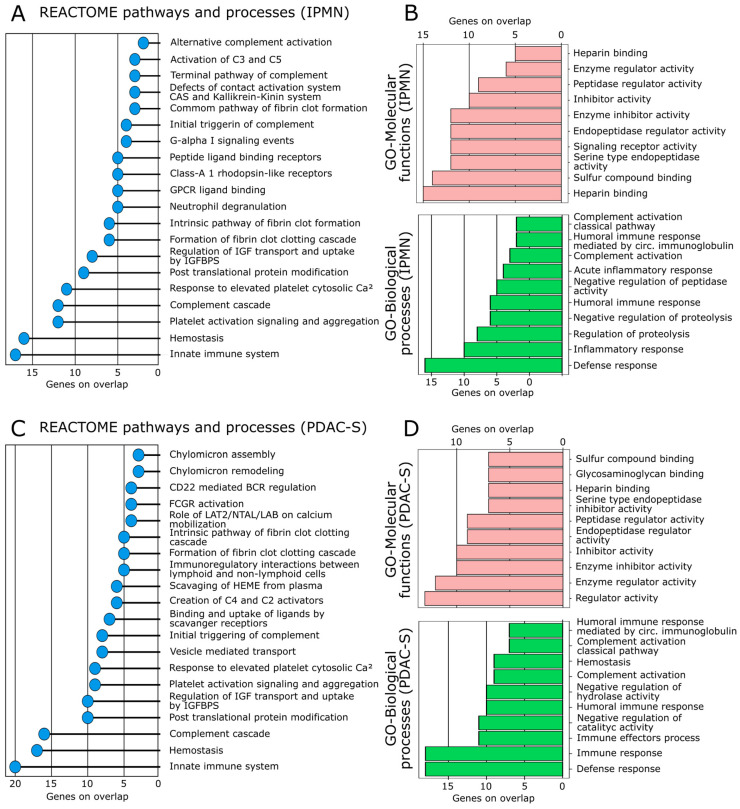
**Biological processes and pathways associated with the IPMN and PDAC-S DEPs.** Pathways and biological processes were determined by the MSigDB (v. 2023.1) database using the REACTOME and Gene ontology GO and MF collections. (**A**,**B**) REACTOME, GO-MF, and GO-BP terms were significantly enriched in the set of IPMN DEPs. (**C**,**D**) REACTOME, GO-MF, and GO-BP terms were significantly enriched in the set of IPMN DEPs.

**Figure 5 ijms-24-12604-f005:**
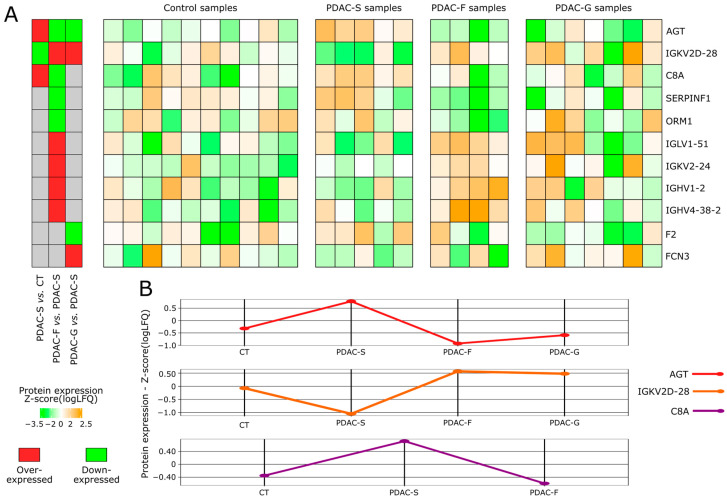
Differentially expressed proteins in PDAC samples after Folfirinox neoadjuvant and Gemcitabine adjuvant therapies. (**A**) Expression pattern of the DEPs identified in the comparison of PDAC-S vs. PDAC-F and PDAC-S vs. PDAC g (*p*-value < 0.05). (**B**) Protein expression alterations were concurrently identified in the PDAC-S vs. PDAC-F, PDAC-S vs. PDAC-G, and PDAC-S vs. CT comparisons. The heatmap indicates the specific comparisons in which the proteins on each line exhibited differential expression. Red: Over-expressed protein. Green: Hypo-expressed protein. Grey: Protein non-differentially expressed in the comparison.

**Table 1 ijms-24-12604-t001:** Epidemiological data for participants with pancreatic cancer and healthy controls for gender, age at diagnosis, tobacco, alcohol usage, and diabetes included in this study.

	Samples	Male	Female	Age (Mean ± SD)	Tobacco (n%)	Alcohol (n%)	Diabetes (n%)
Pancreatic cancer	23	25%	75%	66.08 ± 13.31	9 (39.13%)	5 (21.73%)	9 (39.23%)
PDAC	16	29.50%	70.50%	62.05 ± 11.09	8 (50%)	4 (25%)	4 (25%)
IPMN	7	14.30%	85.70%	75.85 ± 13.93	1 (14.28%)	1 (14.28%)	5 (71.42%)
Healthy control	10	40%	60%	63.7 ± 12.80	1 (10%)	2 (20%)	0 (0%)

**Table 2 ijms-24-12604-t002:** Differentially expressed proteins identified in distinct pancreatic diseases compared to the healthy controls related to cancer drivers.

Gene Symbol	Comparison	Protein Expression Levels *	Candidate Cancer Driver	Reactome Pathways
*APOB*	PDAC-S vs. CT	↓ PDAC-S	Putative tumor suppressor gene	The innate immune system, hemostasis, regulation of insulin-like growth factor IGF transport and uptake by insulin-like growth factor binding proteins IGFBPS, binding and uptake of ligands by scavenger receptors, chylomicron assembly and remodeling, vesicle-mediated transport, and post-translational protein modification
*C3*	IPMN vs. CT, PDAC-S vs. CT	↓ IPMN, ↓ PDAC-S	Putative tumor suppressor gene	Complement pathways, innate immune system, neutrophil degranulation, fibrin clot formation, response to elevated platelet cytosolic Ca^2+^, regulation of insulin-like growth factor IGF transport and uptake by insulin-like growth factor binding proteins IGFBPS, class a 1 rhodopsin-like receptors, GPCR ligand binding, and G alpha I signaling events
*IGLL5*	PDAC-S vs. CT	↑ PDAC-S	Putative oncogene	-
*LBP*	IPMN vs. CT	↑ IPMN	Putative oncogene	Innate immune system
*SERPINA1*	IPMN vs. CT, PDAC-S vs. CT	↑ IPMN, ↑ PDAC-S	Putative tumor suppressor gene	Hemostasis, response to elevated platelet cytosolic Ca^2+^, platelet activation signaling and aggregation, innate immune system, neutrophil degranulation, regulation of insulin-like growth factor IGF transport and uptake by insulin-like growth factor binding proteins IGFBPS, post-translational protein modification, and vesicle-mediated transport
*SERPINA4*	IPMN vs. CT	↑ IPMN	Putative tumor suppressor gene	Hemostasis, response to elevated platelet cytosolic Ca^2+^, platelet activation signaling, and aggregation
*SERPING1*	IPMN vs. CT, PDAC-S vs. CT	↑ IPMN, ↑ PDAC-S	Putative tumor suppressor gene	Complement cascade, hemostasis, response to elevated platelet cytosolic Ca^2+^, platelet activation signaling and aggregation, innate immune system, intrinsic pathway of fibrin clot formation, and clotting cascade

*: Protein expression levels in IPMN and/or PDAC-S compared to the healthy controls. CT—Healthy control patients, IPMN—Intraductal papillary mucinous neoplasms; PDAC—Pancreatic Ductal Adenocarcinoma. ↑ over-expressed. ↓ hypoexpressed.

**Table 3 ijms-24-12604-t003:** Differentially expressed proteins identified in PDAC distinct treatments.

REACTOME Pathways and Processes	PDAC-S vs. PDAC-F	PDAC-S vs. PDAC-G	Gene Symbol
	FDR-Value	FDR-Value	
Complement cascade	5.84 × 10^−6^	1.27 × 10^−4^	C8A, F2, FCN3, IGHV1-2, IGKV2D-28, IGLV1-51
CD22-mediated bcr regulation	6.86 × 10^−5^	-	IGHV1-2, IGKV2D-28, IGLV1-51
FCGR activation	6.86 × 10^−5^	-	IGHV1-2, IGKV2D-28, IGLV1-51
Scavenging of HEME from plasma	6.86 × 10^−5^	-	IGHV1-2, IGKV2D-28, IGLV1-51
Creation of C4 and C2 activators	6.86 × 10^−5^	1.15 × 10^−2^	FCN3, IGHV1-2, IGKV2D-28, IGLV1-51
Role of LAT2/NTAL/LAB on calcium mobilization	6.86 × 10^−5^	-	IGHV1-2, IGKV2D-28, IGLV1-51
Initial triggering of complement	6.93 × 10^−5^	1.15 × 10^−2^	FCN3, IGHV1-2, IGKV2D-28, IGLV1-51
Role of phospholipids in phagocytosis	6.93 × 10^−5^	-	IGHV1-2, IGKV2D-28, IGLV1-51
Antigen activates B cell receptor BCR leading to the generation of second messengers	6.93 × 10^−5^	-	IGHV1-2, IGKV2D-28, IGLV1-51
FceRI mediated Ca²^+^ mobilization	6.93 × 10^−5^	-	IGHV1-2, IGKV2D-28, IGLV1-51
FceRI-mediated MAPK activation	6.93 × 10^−5^	-	IGHV1-2, IGKV2D-28, IGLV1-51
FCGR3A mediated IL10 synthesis	7.94 × 10^−5^	-	IGHV1-2, IGKV2D-28, IGLV1-51
Binding and uptake of ligands by scavenger receptors	7.94 × 10^−5^	-	IGHV1-2, IGKV2D-28, IGLV1-51
Innate immune system	7.94 × 10^−5^	2.99 × 10^−2^	C8A, F2, FCN3, IGHV1-2, IGKV2D-28, IGLV1-51, ORM1
Parasite infection	1.21 × 10^−4^	-	IGHV1-2, IGKV2D-28, IGLV1-51
Anti-inflammatory response favoring leishmania parasite infection	1.73 × 10^−4^	-	IGHV1-2, IGKV2D-28, IGLV1-51
FceRI-mediated Nf-kB activation	1.73 × 10^−4^	-	IGHV1-2, IGKV2D-28, IGLV1-51
FCGamma receptor FCGR-dependent phagocytosis	1.9 × 10^−4^	-	IGHV1-2, IGKV2D-28, IGLV1-51
Potential therapeutics for SART	2.24 × 10^−4^	-	IGHV1-2, IGKV2D-28, IGLV1-51
Signaling by the b cell receptor BCR	2.67 × 10^−4^	-	IGHV1-2, IGKV2D-28, IGLV1-51
Cell surface interactions at the vascular wall	-	3.55 × 10^−2^	F2, IGKV2D-28
Peptide ligand-binding receptors	-	3.55 × 10^−2^	AGT, F2
G-alphaQ signaling events	-	3.65 × 10^−2^	AGT, F2

## Data Availability

No new data were created.

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
