# Peer review of "Screening of Exosome-Derived Proteins and Their Potential as Biomarkers in Diagnostic and Prognostic for Pancreatic Cancer"

_ijms, 2023, doi:10.3390/ijms241612604_

Round 1

Reviewer 1 Report

In this manuscript by Marin et al the authors  analyzed the plasmatic exosome proteomic profile of 23 pancreatic cancer patients and 10 healthy controls by using Nanoscale liquid chromatography coupled to tandem  mass spectrometry (NanoLC-MS/MS). The reviewer feels that overall it is a well written manuscript   which reports important and may be critically important data on novel biomarkers for prognosis and diagnosis of pancreatic cancer. However, the reviewer believes that the manuscript will be improved and the data further corroborated if they can validate some of the exosomal protein biomarker data in mouse pancreatic cancer models.

In this manuscript by Marin et al the authors  analyzed the plasmatic exosome proteomic profile of 23 pancreatic cancer patients and 10 healthy controls by using Nanoscale liquid chromatography coupled to tandem  mass spectrometry (NanoLC-MS/MS). The reviewer feels that overall it is a well written manuscript   which reports important and may be critically important data on novel biomarkers for prognosis and diagnosis of pancreatic cancer. However, the reviewer believes that the manuscript will be improved and the can be data further corroborated if they can validate some of the exosomal protein biomarker data in mouse pancreatic cancer models.

Author Response

Comments and Suggestions for Authors

In this manuscript by Marin et al the authors analyzed the plasmatic exosome proteomic profile of 23 pancreatic cancer patients and 10 healthy controls by using Nanoscale liquid chromatography coupled to tandem mass spectrometry (NanoLC-MS/MS). The reviewer feels that overall it is a well written manuscript which reports important and may be critically important data on novel biomarkers for prognosis and diagnosis of pancreatic cancer. However, the reviewer believes that the manuscript will be improved and the data further corroborated if they can validate some of the exosomal protein biomarker data in mouse pancreatic cancer models.

Comments on the Quality of English Language

In this manuscript by Marin et al the authors analyzed the plasmatic exosome proteomic profile of 23 pancreatic cancer patients and 10 healthy controls by using Nanoscale liquid chromatography coupled to tandem mass spectrometry (NanoLC-MS/MS). The reviewer feels that overall it is a well written manuscript which reports important and may be critically important data on novel biomarkers for prognosis and diagnosis of pancreatic cancer. However, the reviewer believes that the manuscript will be improved and the can be data further corroborated if they can validate some of the exosomal protein biomarker data in mouse pancreatic cancer models.

Answer: We appreciate the reviewer's suggestion and declare that this validation is planned as a future perspective for our group, together with validation in a larger pancreatic cancer cohort.  In this context, we added a last paragraph of future perspectives on the manuscript.

Reviewer 2 Report

Summary:

There is a critical need of blood-based biomarkers for early diagnosis of pancreatic cancer. Authors have performed LC-MS/MS based analysis of proteins derived from plasma exosomes from pancreatic cancer patients and healthy cohort for biomarker analysis. In the study, authors found potential proteins that could be used as biomarkers for diagnostic and prognostic evaluation among healthy and pancreatic cancer groups.

Overall comments:

The field of pancreatic cancer is in the critical need of blood-based biomarkers for diagnostic purposes. Authors have attempted to address this huge unmet need in this study. Overall design of the study is sound. Authors have discussed the relevance of proteins identified in depth in discussion section. However, some of the protein identified may not be specific to pancreatic cancer. Additionally, the number of patient samples in each group might be too low to derive robust biomarkers. There are major comments for protein quantification and statistics methods used in the analysis.

Major comments:

1.       As shown in Figure 1 and described in methods section, proteins derived from exosomes were first separated with SDS-PAGE. Each lane was then further cut into small pieces for in-gel digestion of proteins and subsequent LC-MS/MS analysis. Was there any rationale for separating proteins with SDS-PAGE gel? Proteins derived from exosomes can directly be digested with trypsin. SDS-PAGE can introduce technical variability where variable amounts of proteins could be lost during the process, hindering robust quantification of proteins. Have the authors performed control experiments starting with aliquots of same protein lysate and get reproducible (for example, coefficient of variation < 10% for LFQ) quantification?

2.       Authors used MaxQuant for peptide and protein identifications. It is not clear if the proteins were quantified with at least 2 unique peptides. In biomarker studies, it is crucial to quantify protein levels with at least 2 unique peptides to avoid 1 hit wonders. It’s an accepted norm in the proteomics/mass spectrometry field to quantify proteins with at least 2 unique peptides or multiple PSMs.  

3.       It is not clear if multiple hypothesis testing was performed to evaluate differentially expressed proteins. It looks like Student’s t-tests was applied with nominal P-value cutoff of 0.05 and fold change of 1.5. Since 135 proteins were retained for bioinformatic analysis, it is essential to correct the P-values for multiple hypothesis testing. Either Bonferroni or Benjamini Hochberg based FDR correction should be applied in evaluating differential expression and biomarker identification.

4.       The LFQ values were log2 transformed and then further normalized by Z-score. Was there any rationale for Z-scoring the log2 transformed LFQ values? Z-score can amplify even smaller fold changes if the underlying standard deviation for given protein is a small value. If you perform differential expression analysis on just the log2 transformed values, do you still get same results? Are same set of proteins differentially expressed?

5.       The underlying proteomics data consisting of raw files, etc from mass spec acquisition has been submitted to repository. Authors have provided excel file for differentially expressed proteins. Can authors provide LFQ data (the normalized expression levels and not the z-score) for the all the proteins that were identified in the study in an excel sheet? Such processed data can easily be accessed by readers for any follow-up.   

6.       Patient samples are highly heterogeneous, thus requiring higher number of samples to get reliable biomarkers. Access to patient samples can be limited for exploratory studies like this. Although total of 33 patient samples were analyzed in the study, samples per group might be very few to derive meaningful biological insights.  

7.       Identified proteins such as SERPINAs, APOB, APOC, complement factors (C3, C5, C9, etc) and IgG associated proteins (IGLx, IGKx, IGHx) are some of most abundant proteins in plasma. While the authors attempt to identify proteins in exosomes from plasma, several of these proteins are highly abundant in plasma itself, not just exosomes. With exosome analysis, we would expect to identify more low abundant proteins that play role in signaling cascade such as growth factors or membrane proteins. Could authors comment on the absence of the less abundant proteins.

8.       The differentially expressed proteins between the various groups consist of the same highly abundant proteins. Additionally, several of the proteins such as those involved in complement activation, blood coagulation, lipoproteins, and immune system may not be specific to pancreatic cancer, and can play role in other diseases. Could authors comment on specificity of these biomarkers for diagnostic and prognostic biomarkers.

Can authors comment on predictive ability of differentially expressed proteins in diagnosing pancreatic cancer? 

Minor comments:

1.       Line 227-230: Same sentence is repeated

English language requires minor editing. 

Author Response

Reviewer 2

Comments and Suggestions for Authors

Summary:

There is a critical need of blood-based biomarkers for early diagnosis of pancreatic cancer. Authors have performed LC-MS/MS based analysis of proteins derived from plasma exosomes from pancreatic cancer patients and healthy cohort for biomarker analysis. In the study, authors found potential proteins that could be used as biomarkers for diagnostic and prognostic evaluation among healthy and pancreatic cancer groups.

Overall comments:

The field of pancreatic cancer is in the critical need of blood-based biomarkers for diagnostic purposes. Authors have attempted to address this huge unmet need in this study. Overall design of the study is sound. Authors have discussed the relevance of proteins identified in depth in discussion section. However, some of the protein identified may not be specific to pancreatic cancer. Additionally, the number of patient samples in each group might be too low to derive robust biomarkers. There are major comments for protein quantification and statistics methods used in the analysis.

Major comments:

  1. As shown in Figure 1 and described in methods section, proteins derived from exosomes were first separated with SDS-PAGE. Each lane was then further cut into small pieces for in-gel digestion of proteins and subsequent LC-MS/MS analysis. Was there any rationale for separating proteins with SDS-PAGE gel? Proteins derived from exosomes can directly be digested with trypsin. SDS-PAGE can introduce technical variability where variable amounts of proteins could be lost during the process, hindering robust quantification of proteins. Have the authors performed control experiments starting with aliquots of same protein lysate and get reproducible (for example, coefficient of variation < 10% for LFQ) quantification?

Answer: We agree that in-gel digestion is prone to introduce more variance compared to in-solution digestion. In the core facility, previous attempts to analyze EV protein content after trypsin in solution digestion had failed due to poor peptide recovery. Then, in-gel digestion was successfully implemented for EV analysis. The in-gel digestion protocol used is a well established protocol and has been used for a decade in the core facility. To minimize variances, the SDS solubilized extracts were quantified, and the same protein mass was loaded on the acrylamide gel for all the samples. After desalting, the peptide samples were quantified to assure that the same mass would be injected into LC-MS analysis among the samples. In-gel digestion enhances the clean-up of the samples, dealing with the LC setup of the core facility, in which no pre- or trap-columns are used. The LC-MS system performance was monitored through a regular injection of a cellular protein extract in which 70% of the proteins had a CV < 20% for the LFQ intensity. For future analysis, with a larger sample size, we will adjust our in-solution digestion protocol and consider it for sample preparation. 

  1. Authors used MaxQuant for peptide and protein identifications. It is not clear if the proteins were quantified with at least 2 unique peptides. In biomarker studies, it is crucial to quantify protein levels with at least 2 unique peptides to avoid 1 hit wonders. It’s an accepted norm in the proteomics/mass spectrometry field to quantify proteins with at least 2 unique peptides or multiple PSMs.  

Answer: Although not clearly stated in the manuscript, proteins were only considered for quantification if 2 or more unique peptides were identified for that group. This information was added in line 657-658 in the text. 

  1. It is not clear if multiple hypothesis testing was performed to evaluate differentially expressed proteins. It looks like Student’s t-tests was applied with nominal P-value cutoff of 0.05 and fold change of 1.5. Since 135 proteins were retained for bioinformatic analysis, it is essential to correct the P-values for multiple hypothesis testing. Either Bonferroni or Benjamini Hochberg based FDR correction should be applied in evaluating differential expression and biomarker identification.

Answer: We understand your concerns. Initially, the 1% FDR correction was applied in the spectra identification step, performed in MaxQuant software version 1.6.17.0, as described in item 4.5 of our methodology section. In a second moment, we applied Student’s t-tests to identify the differentially expressed proteins considering the P-value cutoff of 0.05 and fold change of 1.5 (logFC 0.58). The Bonferroni or Benjamini Hochberg based FDR corrections are commonly used approaches to reduce type 1 error when multiple hypothesis tests are performed simultaneously. In our study, we only applied two-sample t-tests and compared paired groups, thereby, not incurring in the issue of multiple hypothesis testing. Our reasoning is based on the discussion of Lee and coworkers, (2018: doi: 10.4097/kja.d.18.00242) and in other proteomic studies that only applied the FDR correction when comparing 3 or more groups simultaneously by an analysis of variance such as ANOVA, for example, but not in two-sample t-tests (GIUDICE et al., 2019; 68: 38–50. doi:10.1016/j.exphem.2018.09.008 and NYLUND et al., 2014; doi: 10.1093/jrr/rru007).  

  1. The LFQ values were log2transformed and then further normalized by Z-score. Was there any rationale for Z-scoring the log2 transformed LFQ values? Z-score can amplify even smaller fold changes if the underlying standard deviation for given protein is a small value. If you perform differential expression analysis on just the log2 transformed values, do you still get same results? Are same set of proteins differentially expressed?

Answer: We appreciate your comments. In our proteomic analysis, we followed the recommended normalization steps as recommended by the Perseus developers. We provided the appropriate references to access the information about the Perseus analysis in the manuscript (4.5 Data analysis). Additionally, we have also utilized alternative normalization methods, such as the width adjustment described in Gomig et al. 2019 (doi:10.1016/j.jprot.2019.02.007). Although our group has frequently employed z-scores in various studies to enable data comparisons, we highly value your comments and we will certainly take it into consideration for our future studies.

  1. The underlying proteomics data consisting of raw files, etc from mass spec acquisition has been submitted to repository. Authors have provided excel file for differentially expressed proteins. Can authors provide LFQ data (the normalized expression levels and not the z-score) for the all the proteins that were identified in the study in an excel sheet? Such processed data can easily be accessed by readers for any follow-up.

Answer: We addressed your suggestion. We added in supplementary table 1 the LFQ intensities of all identified proteins, as well the normalized data of the 135 expressed in at least 70% of the samples.

  1. Patient samples are highly heterogeneous, thus requiring higher number of samples to get reliable biomarkers. Access to patient samples can be limited for exploratory studies like this. Although total of 33 patient samples were analyzed in the study, samples per group might be very few to derive meaningful biological insights.  

Answer: We appreciate your concerns. The mass-spectrometry-based studies often involve smaller sample sets when compared to other omics approaches, such as transcriptomics. The number of patients included in our study was determined by the availability of samples, eligibility criteria for sample selection (diagnosis, treatment and non-treatment), and the types of treatment administered. Thus, our sample size may be a limitation of the study, which was already described in discussion as a weakness (lines 564-569). However, we conducted our analyses aiming to investigate the exosome proteome alterations involved in pancreatic tumorigenesis and stratified a set of PDAC samples according to the neoadjuvant treatment to provide a brief explorative analysis of the possible effects of the gemcitabine and folfirinox in the exosome protein content. All the statistical analyses were based on proteins identified in at least 70% of the samples aiming to provide a more representative proteome. Although larger sample sets are important in research, particularly when studying complex biological systems, our study provides an interesting set of proteins and biological processes that could be altered in pancreatic tumorigenesis. We appreciate your feedback and will consider it in future studies, thereby enhancing the reliability and generalizability of our findings.

  1. Identified proteins such as SERPINAs, APOB, APOC, complement factors (C3, C5, C9, etc) and IgG associated proteins (IGLx, IGKx, IGHx) are some of most abundant proteins in plasma. While the authors attempt to identify proteins in exosomes from plasma, several of these proteins are highly abundant in plasma itself, not just exosomes. With exosome analysis, we would expect to identify more low abundant proteins that play role in signaling cascade such as growth factors or membrane proteins. Could authors comment on the absence of the less abundant proteins.

Answer: We appreciate the comments and agree it is a difficult point in plasma-exossome research. As a first precaution, we used a high quality commercial kit for exosome isolation (miRCURY® Exosome Serum / Plasma Kit - Qiagen). More than that, similar proteins were described in plasma exosomes from pancreatic cancer patients by Yang and colleagues (https://doi.org/10.3389/fonc.2021.628346), including complement and Ig associated proteins. Even in the study of Rebetto and colleagues (10.3390/diagnostics11060917), who used additional methods to exclude contamination of non-EV proteins in plasma, complement, apolipoproteins and globins were found. These proteins are also described in ExoCarta database as presents in human plasma. Further, some plasma proteins have been reported as part of a corona protein that coats plasma EV (https://doi.org/10.1002/jev2.12140). Additional methods of depletion may be useful to exclude most abundant proteins and in this way favor less abundant protein identification, but also could lead to loss of the less abundant ones. 

  1. The differentially expressed proteins between the various groups consist of the same highly abundant proteins. Additionally, several of the proteins such as those involved in complement activation, blood coagulation, lipoproteins, and immune system may not be specific to pancreatic cancer, and can play role in other diseases. Could authors comment on specificity of these biomarkers for diagnostic and prognostic biomarkers.

Answer: We appreciate the comments. We consider that this is a common limitation within other similar studies describing plasma biomarkers in cancer that can only be overcome by: i) the use of appropriate number of control samples. Associated with this, the use of plasma abundant proteins depletion and/or sample fractionation; and ii) the use of new MS instrumentation, for example Thermo Orbitrap Astral. The authors recognize the reduced number of samples in the cohort but emphasize that this is preliminary and exploratory study that aimed to raise candidates for future investigation.

Can authors comment on predictive ability of differentially expressed proteins in diagnosing pancreatic cancer? 

Answer: The heterogeneity in the biology and clinical behavior of pancreatic tumors have driven a joint effort to search for reliable biomarkers to predict the patient´s outcome and disease evolution. In this context, exosome proteins have arisen as non-invasive biomarkers for numerous cancer types. In this research, besides investigating the biological alterations of the IPMN precancerous lesion and PDAC tumors, we propose a list of proteins whose expression can be associated with pathological alterations in pancreas. As differentially expressed, proteins (DEPs) represent the major source of protein biomarkers, altogether with splicing variants and mutant proteins, we believe that our exosome DEPs have potential in liquid biopsy to complement the classic biomarkers and predict the presence of neoplastic alterations in the pancreatic tissue, and also to monitor disease progression after folfirinox and/or gemcitabine treatment. We are aware that validation studies and assessment of specificity and sensibility of our proteins are needed to confirm its potential as biomarkers and consolidate its clinical application, however, our data can pave the way for these studies and guide the researchers to candidate biomarkers. This data provides an initial pathway for larger cohort validation, in which a longitudinal clinical study is in course by our group to validate and provides diagnostic parameters for such DEPs.

Minor comments:

Line 227-230: Same sentence is repeated

Answer: the sentence has been removed.

Round 2

Reviewer 1 Report

Although the manuscript could have been improved, as it stands the results presented in manuscript may be of sufficient importance and recommended for publication in IJMS. 

Only minor editing of English language required

Reviewer 2 Report

Major comments have been addressed by the authors.

Minor editing required